# Targeting of Specialized Metabolites Biosynthetic Enzymes to Membranes and Vesicles by Posttranslational Palmitoylation: A Mechanism of Non-Conventional Traffic and Secretion of Fungal Metabolites

**DOI:** 10.3390/ijms25021224

**Published:** 2024-01-19

**Authors:** Juan F. Martín, Paloma Liras

**Affiliations:** Departamento de Biología Molecular, Área de Microbiología, Universidad de León, 24071 León, Spain; paloma.liras@unileon.es

**Keywords:** subcellular compartmentalization, palmitoylation, secondary metabolism, filamentous fungi, peroxisomes, endosomes

## Abstract

In nature, the formation of specialized (secondary) metabolites is associated with the late stages of fungal development. Enzymes involved in the biosynthesis of secondary metabolites in fungi are located in distinct subcellular compartments including the cytosol, peroxisomes, endosomes, endoplasmic reticulum, different types of vesicles, the plasma membrane and the cell wall space. The enzymes traffic between these subcellular compartments and the secretion through the plasma membrane are still unclear in the biosynthetic processes of most of these metabolites. Recent reports indicate that some of these enzymes initially located in the cytosol are later modified by posttranslational acylation and these modifications may target them to membrane vesicle systems. Many posttranslational modifications play key roles in the enzymatic function of different proteins in the cell. These modifications are very important in the modulation of regulatory proteins, in targeting of proteins, intracellular traffic and metabolites secretion. Particularly interesting are the protein modifications by palmitoylation, prenylation and miristoylation. Palmitoylation is a thiol group-acylation (S-acylation) of proteins by palmitic acid (C16) that is attached to the SH group of a conserved cysteine in proteins. Palmitoylation serves to target acylated proteins to the cytosolic surface of cell membranes, e.g., to the smooth endoplasmic reticulum, whereas the so-called toxisomes are formed in trichothecene biosynthesis. Palmitoylation of the initial enzymes involved in the biosynthesis of melanin serves to target them to endosomes and later to the conidia, whereas other non-palmitoylated laccases are secreted directly by the conventional secretory pathway to the cell wall space where they perform the last step(s) of melanin biosynthesis. Six other enzymes involved in the biosynthesis of endocrosin, gliotoxin and fumitremorgin believed to be cytosolic are also targeted to vesicles, although it is unclear if they are palmitoylated. Bioinformatic analysis suggests that palmitoylation may be frequent in the modification and targeting of polyketide synthetases and non-ribosomal peptide synthetases. The endosomes may integrate other small vesicles with different cargo proteins, forming multivesicular bodies that finally fuse with the plasma membrane during secretion. Another important effect of palmitoylation is that it regulates calcium metabolism by posttranslational modification of the phosphatase calcineurin. Mutants defective in the Akr1 palmitoyl transferase in several fungi are affected in calcium transport and homeostasis, thus impacting on the biosynthesis of calcium-regulated specialized metabolites. The palmitoylation of secondary metabolites biosynthetic enzymes and their temporal distribution respond to the conidiation signaling mechanism. In summary, this posttranslational modification drives the spatial traffic of the biosynthetic enzymes between the subcellular organelles and the plasma membrane. This article reviews the molecular mechanism of palmitoylation and the known fungal palmitoyl transferases. This novel information opens new ways to improve the biosynthesis of the bioactive metabolites and to increase its secretion in fungi.

## 1. Introduction

Filamentous fungi are prolific producers of different classes of specialized metabolites (also named secondary metabolites, used indistinctly in this article), that have important pharmacological activities [1,2]. Since hundreds of species of filamentous fungi have been studied and still many more have not been analyzed, the number of specialized metabolites with putative important activities in fungi will greatly increase. Several reports describe a great diversity in the localization of specialized metabolites biosynthetic enzymes [3]. Increasing evidence suggests that these biosynthetic enzymes are redirected from the cytosol, where they are formed, to vesicles following induction of the production of secondary metabolites that frequently is related with the formation of sclerotia, asexual conidia, sexual ascospores or basidiomycetes fruiting bodies [2,4,5].

In recent decades, there have been several reports on the important role of peroxisomes and endosomes in the localization and traffic of secondary metabolites biosynthetic enzymes and in the export of the final products (see below). Peroxisomes are organelles surrounded by a single membrane. The peroxisomal matrix proteins contain targeting signal sequences of three amino acids with small variations named PTS1 (S/A/C)-(K/R/H)-L) in the C-terminal end and/or PTS2 (R/K)-(L/V/I)-X5-(H/Q)-(L/A) in the N-terminal region, that are recognized by the peroxines Pex5 and Pex7 located in the peroxisomal membrane [6,7,8,9,10,11]. Peroxisomes may perform synthesis and modification/degradation of bioactive metabolites, catabolism of linear or branched-chain fatty acids and removal of H_2_O_2_ by catalase. 

Vesicles are double membrane-surrounded organelles that are formed by budding from peroxisomes, mitochondria, vacuoles or by the plasma membrane invaginations.

## 2. Diverse Localization of Secondary Metabolites Biosynthetic Enzymes

In the following sections, the available information on the subcellular localization and secretion of enzymes involved in the biosynthesis of different classes of bioactive metabolites is reviewed. Particular attention is paid to those fungal model systems in which there is a good characterization of the biosynthetic enzymes of specialized metabolites and a documented knowledge of their localization in different organelles (Table 1 and Table 2).

### 2.1. Penicillins, Cephalosporin C and Cyclosporins

In many filamentous fungi peroxisomes are organelles that harbour secondary metabolite biosynthetic enzymes, at least at certain stages of the growth and differentiation process; e.g., non-ribosomal peptide synthetases (NRPS) in penicillins, cephalosporin C and cyclosporins. Several bioactive metabolites biosynthetic enzymes have been reported to be cytosolic-based mainly on the lack of association with membrane systems or in the intracellular localization of these enzymes labelled with fluorescent tags (e.g., green fluorescent protein, GFP) without further studies on their possible association with fungal organelles; sometimes this has resulted in contradictory reports on the localization of some of these enzymes. This is the case of the α-aminoadipyl-cisteinyl-valine (ACV) synthetase, an NRPS that catalyses the first step in the biosynthesis of the beta lactam antibiotics penicillins and cephalosporin C [35]. Studies on the localization of the first two enzymes of the penicillin biosynthesis pathway, namely ACV synthetase and isopenicillin N synthase, isolated by sucrose gradient ultracentrifugation, indicated that the ACV synthetase, but not the isopenicillin N synthase, is associated with membrane systems [36]. However, these results were disputed by Van der Lende et al. (2002) [37], who, using electron microscopy and electron immunofluorescence, showed that both these enzymes were localized in the cytosol. It is important to note that this discrepancy is not contradictory since it is possible that the ACV synthetase is first located in the cytosol and later targeted to vesicles [38,39]. The biosynthetic enzymes usually found in the cytosol lack secretion signal peptides or transmembrane domains for their association/localization in membranes. However, some enzyme modifications, such as lipid attachment (lipidation), may target them to vesicles or endosomes that may serve as transport systems in intracellular traffic and secretion (see Section 4 below). 

Peroxisomal enzymes include the last two enzymes of the penicillin biosynthesis pathway: (1) The isopenicillin N acyl transferase, IAT [40,41,42] and (2) The phenylacetyl-CoA ligase [12,43,44]. These two enzymes have canonical PTS1 peroxisomal targeting sequences (ARL; SKI or AKL, respectively) in the C-terminal end. Later studies using immuno-microscopy and gold labelling electron microscopy confirmed unequivocally that the isopenicillin N acyl transferase is located in peroxisomes [6,13,45]. *Penicillium chrysogenum* mutants defective in peroxisomes are unable to synthesize penicillin and do not show detectable levels of IAT in contrast to the peroxisome defective mutants of *Aspergillus nidulans* that still maintain a small level of IAT in the cytosol and produce residual amounts of penicillin [10,46] (Figure 1). The transport steps that secrete benzylpenicillin from the peroxisomes to the extracellular medium are still poorly known, although the involvement of multivesicular bodies in the secretion has been proposed [38,39]. The involvement of vesicles in the penicillin secretion process is supported by findings showing that induction of penicillin biosynthesis by diamines results in a significant stimulation of the formation of punctated vesicles, although the nature of these vesicles has not been studied [47].

In *Acremonium chrysogenum*, two enzymes involved in the conversion of isopenicillin N to penicillin N, namely isopenicillin N acyl CoA ligase and isopenicillinyl CoA epimerase, are located in peroxisomes [8,48].

Another example of NRPS linked to the vacuolal membrane is the cyclosporin synthetase of *Tolypocladium inflatum* that synthesizes the potent immunosuppressor cyclosporin. This enzyme is an unusually large NRPS consisting of 11 modules that activate and condense several proteinogenic and other rare non-proteinogenic amino acids. One of the rare amino acids is D-alanine, which is formed from L-alanine by a dedicated alanine racemase, that is non-integrated in the NRPS. Localization studies of the cyclosporine synthetase, encoded by *simA*, showed that it is closely linked to the alanine racemase, forming a punctated structure that seems to correspond to eleven modules connected by linker regions; both enzymes are attached to the vacuole membrane, suggesting that this organelle may contribute to the formation of the large non-ribosomal peptide [49]. Related to the vacuolal membrane attachment of NRPSs is the finding of a vacuolal transporter, PenV, known to be involved in the ACV tripeptide formation in the biosynthesis of penicillin [50].

### 2.2. Aflatoxins: Integration in Endosomes of Enzymes from Distinct Subcellular Compartments

The biosynthesis of aflatoxins has been intensively studied in *Aspergillus parasiticus*, *Aspergillus flavus* and in the sterigmatocystin producer *A. nidulans*. Peroxisomes were proposed to be the site for the NorS complex (PKS-A/AflA/AflB) of the biosynthesis of aflatoxin [51] based on the observation that peroxisomes provide acetyl-CoA, a precursor for polyketides, fatty acids in *A. nidulans* and early intermediates of aflatoxins [52], although this requires further confirmation. Support for this hypothesis was the finding that norsolorinic acid, the product of the first step/s of aflatoxin biosynthesis, is found in peroxisomes where it co-localizes with typical peroxisomal enzymes such as isocitrate lyase. It is still unclear if norsolorinic acid is synthesized in peroxisomes or whether it is formed in the cytosol or mitochondria and introduced into peroxisomes; for comparative purposes, during the biosynthesis of penicillin in *P. chrysogenum*, the preformed phenylacetic acid is introduced in peroxisomes from the cytosol [53]. Several aflatoxin biosynthetic enzymes in *A. parasiticus* including Nor-1, Ver-1 and OmtA are known to be cytosolic, but in the late steps of the producer fungus development they are located in endosomes [26,27,28,51,54]. Later evidence indicated that these vesicles, termed aflatoxisomes, are the site for the final steps of aflatoxins biosynthesis and seem to be fused to the plasma membrane during the secretion process [55,56,57,58]. In addition, another enzyme of the aflatoxin pathway, versicolorin B synthase, is described to be transported via the endoplasmic reticulum conventional secretory pathway to vesicles [54].

Another protein, AflJ, encoded by a gene in the aflatoxin gene cluster [15,59], is also located in endosomes. AflJ contains three membrane-spanning domains and is encoded by a gene divergent from *aflR* that encodes the Znc^2+^-cys6 master regulator of aflatoxin biosynthesis in *A. parasiticus.* Early studies indicated that AflJ is targeted to microbodies (peroxisomes), and indeed both AflJ and the homologous protein in *A. nidulans*, AfgJ, contain in the C-terminal a putative PTS1 peroxisomal targeting sequence [16]. However, additional studies showed that AflJ co-localizes with the regulatory protein AflR in endosomes and also in the periphery of the nucleus, suggesting that the AflJ initially targeted to peroxisomes migrates to the cytosol and endosomes [16]. These authors propose that AflJ helps AflR in its transit from the cytosol into the nucleus, thus controlling the possibility of AflR to enter in the nucleus and activate the entire aflatoxin gene cluster. The AflJ protein does not bind AflR recognition sites excluding the possibility of a direct transcriptional effect similar to that of AflR. Later, using a mutant defective in the AflJ gene, it was observed that endosomes of the mutant strain are retained in the cytosol or near the septa and it was proposed that AflJ may also be involved in helping targeting the endosomes to the cell membrane for export. The question of how the AflJ protein moves from microbodies to endosomes needs further clarification [16,51,60]. It is known that peroxisomes migrate together with endosomes along the hyphae of filamentous fungi and this contact between organelles may cause the transfer of peroxisome enzymes to endosomes [61] (see Section 3 below). 

### 2.3. The Alternaria Alternata AK Toxin Family: A Model of Polyketide Biosynthesis in Peroxisomes

*Alternaria alternata* is a plant pathogen that produces necrosis lesions in the leaves of several fruit plants [62]. Variants of *A. alternata* produce black spots in Japanese pear trees (named AK toxin), strawberries (AF toxin) and tangerines (ACT toxin). All three toxins contain a common moiety of 9, 10 epoxy, 8 hydroxy, 9 methyl-decatrienoic acid (EDA). Early studies on the biosynthesis of the AK toxin (AKT) identified four genes, *AK1*, *AK2*, *AK3* and *AKR*, involved in this biosynthetic pathway [17,18,19]. The proteins encoded by these genes correspond to an acyl-CoA activating enzyme (AK1), an α-β hydrolase (AK2), an enoylhydratase/epimerase (AK3) and a transcriptional regulator (AKR) of the Zn (II)2-Cys6 class. Later studies identified additional genes involved in the biosynthesis of AKT, namely a highly reducing polyketide synthase, a hydroxymethyl glutaryl-CoA synthetase (AK4) and a P450 monooxygenase (AK7) [18,19]. The AK1, 2 and 3 proteins contain C-terminal tripeptides (SKI, SKL, and PKL, respectively), characteristic of the peroxisomal PST1 sequence [9]. The three proteins tagged with GFP were shown to be located in organelles that were identified as peroxisomes [9]. Mutants of *A. alternata* defective in the Pex6 peroxin, a peroxisomal membrane protein required for import of matrix proteins in these organelles, lacked production of AK toxin. These mutants were unable to form peroxisomes, and therefore lack the biosynthesis of EDA, and were unable to infect the host plant [7,63]. Collectively, the available information indicates that the EDA moiety of the AK toxins derives from a long chain polyketide encoded by a high reducing PKS that is modified by the AK1, AK2 and AK3 enzymes. The assembly of the final AK toxins from EDA and other precursors is still unclear. In summary, the formation in peroxisomes of a long chain polyketide such as that involved in the biosynthesis of EDA and its modification by ancillary enzymes, also located in peroxisomes, provides a good model of the involvement of peroxisomes in the formation of polyketide-derived bioactive metabolites.

### 2.4. Peroxisomal Enzymes Involved in the Biosynthesis of Fusarinine Siderophores

Siderophores play a key function in the iron metabolism including scavenging of this metal from different habitats, transport and metabolism, in fungi. The major group of siderophores corresponds to fusarinine and triacetylfusarinine. The structure of fusarinine consists of three N5-anhydromevalonyl-N5-hydroxyornithine molecules linked in a cyclic form by ester bonds. The biosynthesis of this precursor starts with the hydroxylation of ornithine by the N5-ornithinehydroxylase encoded by the *sidA* gene [64,65,66]. The second half of the N5-anhydromevalonyl-N5-hydroxyornithine precursor is formed from mevalonate, which is activated with CoA by SidI and then dehydrated by the SidH enzyme, respectively [67]. The anhydromevalonyl-CoA precursor transfers its anhydromevalonic moiety to N5-hydroxyornithine to form N5-anhydromevalonyl-N5-hydroxyornithine by a ligase encoded by *sidF* [65]. and finally, three units of this intermediates are condensed to form fusarinine C by an NRPS-like protein encoded by the *sidD* gene [25]. The three enzymes leading to the formation of triacetylfusarinine, SidF, SiH and Sid I, have peroxisomal targeting signals are in the C-terminal end (SidF, SidH) or in the N-terminal region (SidI). Removal of the C-terminal sequence in SidH showed that this protein does no enter in peroxisomes and the mutant does not produce triacetylfusarinine. The protein SidH was demonstrated to be located in peroxisomes by labelling the enzyme with fluorescent tags that co-localizes with peroxisomal markers [25]. These results demonstrate that these enzymes accumulate intermediates or final products in peroxisomes.

### 2.5. Trichothecenes: Localization in Toxisomes, a Novel Concept Involving Vesicles Fusions

*Fusarium graminearum* produces a very severe disease of wheat, barley, rice and other cereals named Fusarium head blight disease [68,69,70,71]. The trichothecene, deoxynorvalinol (DON), is one of the most potent mycotoxins produced by different strains of the *F. graminearum* subspecies complex [72]. 

The gene cluster for trichothecenes biosynthesis in *F. graminearum* contains 16 genes including essential and some non-essential genes [73,74,75]. The trichothecenes derive from farnesyl-pyrophosphate, a compound common to the primary metabolism; farnesyl-PP is formed from mevalonic acid that derives from 3-hydroxy-3-methylglutaric acid (HMG) by the action of the HMG-CoA synthetase and the HMG-CoA reductase (Hmr1). Farnesyl-PP is the direct precursor of trichodiene, the first intermediate of the trichothecene pathway, formed by the trichodiene synthase (Tri5). This compound is converted in trichothecene by the action of two P450 oxygenases among other enzymes. 

Early evidence indicates that trichothecene biosynthesis takes place in endosomes (named toxisomes) [30]. It is interesting that once the endosomes are formed during the stationary growth phase, association of endosomes and peroxisomes has been observed microscopically in *F. graminearum* [31], however, fluorescence labelled peroxisomes do not overlap with endosomes indicating that at that stage both organelles are separated. 

The early enzyme, HMGR, is an integral membrane protein localized at the endoplasmic reticulum in *F. graminearum* as shown by Hmr1-tagging with GFP and by co-purification with other proteins of the endoplasmic reticulum [29]. The two trichothecene biosynthetic cytochrome P450 monooxygenases, named trichodiene oxygenase and calonectrin oxygenase (Tri1 and Tri4) labelled with GFP were shown to co-localize in vesicles (toxisomes) that appear to be the site of the final steps of trichothecenes biosynthesis. 

Interestingly, the HMG-CoA reductase involved in the first step of isoprenoid and trichothecene biosynthesis, together with other trichothecene biosynthesis enzymes, changes its localization following induction of trichothecenes formation from endomembrane systems to vesicles (toxisomes). Labelled HMG-CoA reductase, is localized in the periphery of organelles similar to toxisomes that increase in number following the induction of trichothecene biosynthesis [29]. These authors suggest that the toxisomes containing the HMG-CoA reductase are formed by incorporation of this enzyme to preformed endosomes [30,76]. These results indicate that the structural development of new vesicles is linked to the induction of trichothecene biosynthesis [29]. This response to secondary metabolism inducers agrees with the observation of the increase on punctated vesicles in *P. chrysogenum* following the induction of penicillin biosynthesis by diamines and provides a fungal model for the coordinated induction of subcellular traffic. Experimental data on the mechanisms and signals that direct the trichothecene biosynthesis enzymes to the toxisomes is still needed; these mechanisms may involve posttranslational modification by S-acylation as shown in the targeting of enzymes involved in the biosynthesis of melanin (see below Section 5). 

Finally, Tri12, a 12 transmembrane domain transporter, known to be involved in detoxification and export of trichothecenes was also localized in these vesicles [29]. This localization raises the question of the function of this Tri12 transporter. What is the role of membrane facilitator superfamily (MFS) proteins in the transport and secretion of fungal metabolites? It was believed that the Tri12 protein was a membrane transporter for secretion of DON and other trichothecenes out of the cells [75] but Tri12 has been reported to be present in vesicle membranes and may serve as the interacting membrane patch of toxisomes thus facilitating the fusion with the plasma membrane and secretion in the producer fungi. Based on these results and in the identification in β-lactam producing fungi of several peroxisomal MFS transporters that also contain 12 or 14 transmembrane domains, e.g., PenM and PaaT in penicillin biosynthesis [53,76] and CefM and CefP [14,77] in cephalosporin C biosynthesis, we propose that many MFS membrane transporters identified in different fungal gene clusters are located in the endosomal or peroxisomal membranes and have a role in the introduction of preformed specialized metabolites precursors or intermediates in these organelles. 

In summary, a new concept of toxisomes is evolving in which these organelles are formed by fusion of vesicles of different origins thus serving to integrate different precursor units in the biosynthesis of complex secondary metabolites.

### 2.6. Mycophenolic Acid: Diverse Localization of the Biosynthetic Enzymes

Mycophenolic acid, produced by a few *Penicillium* species, is an immunosuppressant and has antibiotic activity against *Bacillus* species, e.g., *Bacillus anthracis.* The gene cluster for mycophenolic acid biosynthesis was first cloned from *Penicillium brevicompactum* and later from *Penicillium roqueforti* [78,79]. The cluster contains seven genes encoding the proteins MpaC, for a non-reductive iterative PKS-methyltransferase required for the synthesis of the intermediate 5-methylorsellinic acid from one starter acetyl-CoA, three malonyl-CoA extender units and a methyl group, provided by an integrated methylation domain of the PKS. The other genes are *mpaA*, encoding a prenyltransferase; *mpaB*, encoding an oxygenase for cleavage of the farnesyl side chain; *mpaDE*, encoding a bifunctional protein corresponding to a fused P450 monooxygenase and a hydrolase; *mpaH* an acyl-Coa hydrolase, *mpaG*, for a stand-alone O-methyltransferase [80] and *mpaB* encoding an oxidative cleavage enzyme. In addition, *mpaF*, encodes an inosine-5-phosphate dehydrogenase that contributes to the resistance to mycophenolic acid. The function of the proteins has been determined by heterologous expression in *A. nidulans*, enzymatic assays, feeding precursors and conversion analysis, protein labelling with fluorescent tags and fluorescence microscopic observation [20]. The biosynthesis of mycophenolic acid starts with the formation of a tetraketide by the polyketide synthase MpaC to produce 5-methyl-orsenillic acid (MOA) that is cyclized by MpaD to form 4-farnesyl-3,5-dihydroxy-6-methylphthalide (FDHMP). Zhang et al. [20] confirmed that the intermediate FDHMP is prenylated by the *mpaA* encoded prenyltransferase that attaches the C-15 farnesyl group to C-4 forming the 4-farnesyl derived compound (FDHMP). This intermediate is extracted only from the mycelium but not from the medium supernatant since the high molecular weight of this prenylated compound prevents its transport through the cell membrane. MpaB is a cleavage enzyme that attacks the C15–C16 double bond of the substrate removing a three-carbon side chain of MDPF what forms the 21 carbon MDPF-3C intermediate. Finally, the action of the O-methyltransferase MpaG results in the formation of mycophenolic acid. 

Zhang et al. [20] established that the lineal side chain in mycophenolic acid is formed by the fatty acid degradation system through the β-oxidation mechanism and were able to isolate and identify a peroxisomal acyl-CoA ligase (PbACL891) that activates the intermediate to start the β-oxidation process and an acyl-CoA hydrolase (MpaH) that removes the CoA group. The subcellular localization of the mycophenolic acid biosynthesis enzymes in *P. brevicompactum* is diverse: the initial PKS MpaC and the O-methyltransferase MpaG are located in the cytosol; the localization of the prenyltransferase MpaA is in the Golgi; the MpaB oxygenase and MpaDE fusion protein are located in the endoplasmic reticulum; finally, the long chain fatty acid acyl-CoA ligase and the acyl-CoA hydrolase are peroxisomal enzymes. In vitro studies confirm that this acyl-CoA hydrolase is a type I thioesterase that removes the CoA group from the side chain thus preventing formation of degradation products favouring accumulation of mycophenolic acid with the adequate length (7 carbons) of the lineal chain. The MpaH enzyme is also located in peroxisomes and contains the peroxisomal targeting sequence glycine-lysine-leucine (GKL) and, indeed, this location was confirmed by florescence microscopy of the labelled enzyme. In conclusion, the prenylation reaction in mycophenolic acid biosynthesis occurs in the Golgi apparatus as described for protein prenylations in other microbial systems whereas the initial PKS is located in the cytosol and the fatty acid activation and processing enzymes occur in peroxisomes This elaborated distribution of the mycophenolic acid biosynthetic enzymes highlights the complexity of the subcellular localization of enzymes that include PKSs, prenyltransferases and fatty acid degradation systems, and points out to the need of coordination mechanisms to provide the adequate concentration of intermediates for optimal production of mycophenolic acid [20]. 

### 2.7. Peptidyl Alkaloids: An Extracellular Biosynthetic Enzyme

Another interesting example of different subcellular localization is that of the enzymes for fumiquinazolin C biosynthesis in *Aspergillus fumigatus*, a compound that belongs to a family of peptidyl alkaloids. Fumiquinazolin C is formed by four enzymes encoded in a cluster formed by genes: *fmqA* to *fmqD*. The product of the first gene of the cluster, *fmqA*, is a tri-modular NRPS that activates and condenses anthranilate, tryptophan and alanine, and is located in vesicles in *A. fumigatus* [32,33], although the exact nature of these vesicles is still unclear. The second enzyme, an oxidoreductase encoded by *fmqB* and the third enzyme, FmqC, a mono-modular NRPS that activates an additional molecule of alanine, are located in the cytosol [34,81]. Surprisingly, the enzyme for the last step of the pathway, FmqD, is a cell-wall associate extracellular enzyme that converts the final intermediate of the pathway into fumiquinazoline C [82]. This final enzyme is located in the conidial cell wall matrix, and is secreted by the conventional secretory pathway. This provides an interesting example of extracellular localization of the last enzymes of biosynthesis of secondary metabolites that may serve to avoid the toxicity of the final product to the producer strain.

### 2.8. Melanin: The Second Half of the Pathway in Botrytis Cinerea Is Extracellular

Melanin is an important dark pigment that protect animal cells and also the producer fungi against UV irradiation damage, reactive oxygen species and other stressing environmental factors. These pigments also serve as virulence factors used by the pathogenic fungi to attack their hosts [83,84]. Melanin is formed by polymerization of phenolic or indolic intermediates, synthesized by PKSs. This pigment is produced by many filamentous fungi using two different pathways and precursors: they derive from 1,8 dihydroxynaphtalene (DHN) or from 3,5 dihydroxyphenylalanine (L-DOPA), a precursor that is common in animal cells but is rarely used in filamentous fungi [85]. Most ascomycetes use the DHN melanin pathway but there are a few examples also of the L-DOPA pathway, e.g., the basidiomycete *Cryptococcus neoformans* use L-DOPA as precursor and forms melanin in vesicles that are similar to animal melanosomes [83,84]. Extracellular DHN-derived melanins have been studied in several fungi [86,87] and in the last few years significant advances have been made on the full characterization of the pathway and the localization of the biosynthetic enzymes in *A. fumigatus* [85], *B. cinerea* [21,87] and *Neurospora crassa* [88].

There are some differences in the DHN-derived biosynthetic pathways in these three fungi that have been characterized; the designation of the biosynthetic enzymes in these three fungi is different and to avoid confusion the names given by the authors to the respective melanin biosynthetic enzymes are used here. 

In *N. crassa* the DHN melanin is synthesized from acetyl-CoA and malonyl-CoA by a PKS that forms a large polyketide (heptaketide) that is trimmed down to the cyclic intermediate 1,3,6,8 tetrahydroxynaphtalene (THN) and converted to 1,8 dihydroxynaphthalene (DHN) by the THN reductase and the hydrolases PKH1 and PKH2. The DHN molecules are secreted and activated at the cell wall level by the laccase (Lac1) that forms a DHN-free radical that is finally polymerized by Lac1 using oxygen for the polymerization reaction [88].

In *A. fumigatus* six enzymes are encoded in the melanin gene cluster. The first of these enzymes is the polyketide synthase, Alb1, that together with the chain shortening enzyme Ayg1, and the modifying Arp1 and Arp2 forms the DHN intermediate [89,90]. The Ayg1, Arp1 and Arp2 enzymes do not have the conventional leader peptide for secretion and also lack transmembrane domains that might result in their insertion in vesicles or the plasma membrane. Regarding the subcellular localization the chain shortening hydrolase Ayg1 and the early enzymes Arp1 and Arp2 (scytalone dehydratase and 1,3,6,8 THN reductase) are all located in endosomes. The melanin polymerization process is completed by the two laccases and Abr2 [22]. Localization studies showed that these laccases are present in the cell wall in the periphery of the conidia and is known that they are secreted through the conventional secretory pathway. 

Significant advances have been made recently in our understanding of the traffic of intermediates and localization of the enzymes of melanin biosynthesis in a different fungus, the plant pathogen *B. cinerea.* Noteworthy, in *B. cinerea* melanin is synthesized by a six enzymes system which is slightly different from that in *A. fumigatus*, particularly in the localization of the enzymes. *B. cinerea*, synthesize 1,8 DHN starting from a polyketide encoded either by the BcPKS12 or BcPKS13 polyketide synthases (in sclerotia or conidia melanogenesis, respectively) in association with the polyketide trimming down hydrolase BcYGH1 that form THN. This compound is then reduced by the BcBRN1 reductase to scytalone which is secreted. The scytalone is dehydrated to 1,3,8-trihydroxynaphthalene by the BcSCD1 enzyme and this intermediate is reduced to vermelone by the BcBRN1/2 reductases. Finally, this compound is dehydrated by BcSCD1 to form 1,8 DHN that polymerizes to the amorphous melanin complexed with protein and saccharides as granule deposits in the cell wall [86] (Figure 2).

Mutant studies showed that the intermediate scytalone is secreted and accumulated extracellularly in the *B. cinerea* strain Bcsdc1; further observations using labelled proteins stablished that the first enzymes of the pathway up to the formation of scytalone are intracellular whereas the conversion of scytalone to melanin takes place in the cell wall space. Interestingly the first two enzymes of the pathway, BcPKS13 and BcYGH1, are located in peroxisomes since the labelled proteins co-localize with peroxisomal fluorescent labelled markers. Indeed, the carboxy terminal end of PKS12 and PKS13 contain the sequences LTM and LKM that are putative peroxisomal Pts1 targeting signals [7,8]. The reductases BcBRN1/2 were found in endosomes together with the scytalone dehydratase BcSCD1 on its way to the cell wall where they are finally located [21]. Removal of the cell wall, by treatment with lytic enzymes to form protoplasts, released the cell wall associated last enzymes to the culture broth therefore preventing the conversion of scytalone to melanin; when protoplasts were regenerated the cells regain the capability to convert scytalone to melanin [21]. The accumulation of scytalone exerts a self-inhibitory effect on sporulation of *B. cinerea* and this suggests that intracellular compartmentalization and secretion of scytalone prevents its toxic effect. These results provide support to the hypothesis of Cundliffe and Demain (2010) [91] proposing that a mechanism to avoid the toxicity and suicide of the producer strains is to export the toxic product. Whether the secretion and extracellular accumulation of scytalone occurs in other filamentous fungi or if this is a characteristic feature of *B. cinereus* strains remains to be clarified.

In summary, Chen et al. in 2021 [21] concluded that the biosynthesis of melanin is separated into two parts: in the first part the enzymes are located in peroxisomes and endosomes where formation of scytalone is catalyzed whereas in the second part of the pathway the conversion of scytalone to melanin is entirely extracellular. This proposal agrees with the localization of the scytalone dehydratase and the Lac1 laccase in the wall. Genes for other laccases occur also in the genome of *B. cinerea* but the encoded enzymes do not appear to be involved in the melanin polymerization process.

The notable differences in melanin biosynthesis enzymes and their localization in *A. fumigatus* and *B. cinerea* evidence that the secondary metabolite biosynthetic processes and the enzymes localization show a great plasticity in different fungi (Figure 2). Differences may be due to distinct targeting and traffic mechanisms guided by posttranslational modification of the corresponding enzymes, e.g., by palmitoylation. The diversity in melanin biosynthesis is even greater in those fungi that use the L-DOPA pathway as is the case of the basidiomycete *C. neoformans* that contains melanosomes; these are elaborated organelles that also serves as stores of calcium and polyphosphate [84].

### 2.9. Characteristic Features of Enzymes Involved in the Biosynthesis of Other Less Known Specialized Metabolites

The previous sections describe well-known biosynthetic enzymes and their localization, since it provides a global view of the formation of the enzymes, of the intracellular traffic and of their secretion in different fungi. In other secondary metabolite biosynthetic pathways only one or a few enzymes have been characterized [92], but some of the described features are relevant for a better understanding of the localization and secretion of these specialized metabolites. These features include the presence of duplicated enzymes in the biosynthesis of paxilline and the localization of different types of transporters in the biosynthesis of patulin or viriditoxine, respectively.

The indol diterpenoid paxilline is produced by *Penicillium paxilli* and some other fungi. The core of this secondary metabolite is formed by a molecule of geranylgeranyl pyrophosphate (GGPP) that is synthesized by two isoenzymes, namely GGPP synthase A and B. Years ago it was proposed that one of these GGPP synthases was involved in the biosynthesis of primary isoprenoids and the other could be an enzyme for the biosynthesis of paxilline [93]. Recently it was demonstrated that the GGPP synthase B, encoded by the gen *paxG*, was involved in the biosynthesis of paxilline. The *paxG* encoded enzyme contains the PTS1 tripeptide C-terminal sequence and the protein labelled with fluorescent tags was shown to co-localize with peroxisomes. Mutants lacking the C-terminal region fail to synthesize paxilline and the enzyme could not enter into the peroxisomes. Interestingly, this mutation was not complemented by the GGPP synthase A, that was located in vesicles [24]. confirming the hypothesis that two GGPP synthases have different functions in the producer fungus. In summary, it seems that a duplicated GGPP synthase involved in the biosynthesis of isoprenoids in primary metabolism has evolved adapting its localization in peroxisomes to synthesize a GGPP molecule specific for paxilline biosynthesis.

Another relevant feature is the co-localization of the transporters involved in the biosynthesis of patulin in *Penicillium expansum.* The biosynthesis of patulin proceeds from a tetraketide, through 6 methyl salicylic acid and gentisyl alcohol [94]. Most of the enzymes involved in the biosynthesis of patulin are cytosolic or associated with the endoplasmic reticulum. Of the three patulin cluster transporters PatC is an MFS protein with 12 or 13 transmembrane domains whereas PatM is an ABC transporter. The PatA transporter is a small protein with six transmembrane domains identified as a acetate transporter [95]. The two transporters PatC and PatM were found to be located in the plasma membrane while the acetate PatA transporter was located in the endoplasmic reticulum [96]. In the biosynthesis of viriditoxin, a potent toxin produced by *Paecilomyces variotti* the early enzyme PKS and other enzymes for viriditoxin biosynthesis were found to be located in vesicles different from peroxisomes but not fully characterized. In the viriditoxin cluster there is a transporter named VdtG. This transporter is an MFS protein of 12 or 13 transmembrane domains located in branched proliferations of the smooth endoplasmic reticulum, probably coinciding with the formation of toxisomes [97]. The association of VdtG with the formation of toxisomes provides support to the hypothesis that some of the MFS transporters present in secondary metabolites biosynthetic gene clusters may be directly located in the membrane of endosomes and/or toxisomes [97]. These results demonstrate that these transporters accumulate intermediates or final products in peroxisomes that need to be transferred either to the extracellular space or to vesicles; these organelles are responsible for this traffic from the peroxisomes to the plasma membrane.

In summary, the localization of several transmembrane transporters in distinct membrane systems and particularly in the endoplasmic reticulum indicates that these transporters have a critical function in the incorporation of precursor/intermediates into vesicles as described above (Section 2.5).

## 3. Cytosolic Traffic of Peroxisomes, Vesicles and Endosomes: Peroxisomes Move to the Hyphal Tips by Hitchhiking on Early Endosomes

In the cytosol of fungal hyphae there is an important movement of proteins and small molecules, organelles and different types of vesicles. The role of the cytosol is similar to that of a traffic exchanger platform. Peroxisomes are known to be targeted to the hyphal tips where they play an important role in penicillin and other secondary metabolites biosynthesis [98,99]. Movement of peroxisomes and other internal organelles and protein complexes occurs in the hyphal tips along microtubule tracks that are polarized filamentous structures in which the positive end is close to the hyphal tip whereas the negative end lies in the periphery of the nucleus [100,101,102]. The intracellular movement is powered by the dynein and kinesin motors; the first moves the cargo proteins towards the negative end whereas the kinesin moves it to the positive end [103,104]. Salogiannis et al. [61]. observed that the normal distribution and movement of peroxisomes in the hyphae of *Aspergillus nidulans* occurs by hitchhiking on early endosomes. This requires a peroxisome distribution protein (PxdA) that in the wild type strain normally takes care of the peroxisome distribution along the hyphal cells. This protein acts as a coupling agent that tethers peroxisomes on to the early endosomes. Labelling studies and time laps imaging show clearly that labelled peroxisomes localize to the front edge of early endosomes and travel together with endosomes to the hyphal tips. The PxdA protein recruit kinesin and dinein motors for the movement. Mutants lacking PxdA protein do not distribute evenly the peroxisomes in contrast to the parental wild type strain. Analysis of the PxdA protein reveals that it contains a coiled coil domain in the central region of the protein that is essential and sufficient for hitchhiking on the endosomes. An interesting question is how the peroxisomal cargo proteins, that includes several secondary metabolites biosynthetic enzymes, are transferred from the peroxisomes to the cell membrane perhaps by budding of specialized vesicles [61]. This subject requires additional experimental work.

## 4. S-Acylation of Proteins: Palmitoylation

Many posttranslational modifications play key roles in the enzymatic function of different proteins. These modifications include O-acylations, phosphorylations, glycosylations, methylations, prenylations and lipid S-acylations (lipidations), among others. In bacteria, in fungi, as well as in higher eukaryotic cells, these modifications are very important in the control of regulatory proteins such as two component systems [105] and also in the targeting and intracellular traffic of proteins and their secretion [106]. Particularly interesting are the protein modifications by isoprenoids or fatty acids. Palmitoylation, prenylation and miristoylation serve to target the substrate proteins to the cytosolic surface of membranes thus allowing integration in specific sites of vesicles or the plasma membrane [107,108,109]. Palmitoylation is an S-acylation of proteins by palmitic acid (C16) that is attached to the thiol group of a conserved cysteine in proteins and plays a key function in membrane protein traffic and cell signaling. Palmitoylation differs from prenylation and miristoylation in that it frequently acylates integral membrane proteins; these include many G-protein coupled receptors (GPCRs) that play very important roles in nutrient sensing and transport [110], ion channels and neurotransmitters [111]. Also, palmitoylation affects proteins by regulating membrane attachment and integration in different membrane small patches. Protein palmitoylation is an easily reversible process: in the forward direction proteins are acylated by palmitoyl acyltransferases (PAT) and in the reverse direction the palmitoyl chain may be removed by depalmitoylases or some thiosterases. This reversibility favours the rapid adaptation of protein traffic and protein localization as required by the cells. Some palmitoyl transferases use different fatty acids for protein modification, for example stearic acid (C18) and unsaturated long chain fatty acids [112,113].

### 4.1. Palmitoylation of Proteins in Yeasts

Early studies identified seven palmitoyl transferases in *Saccharomyces cerevisiae* [108,114,115]. These include the well characterized PATs Akr1, Erf2-Shr5 and Swf1. The palmitoylated proteins in *S. cerevisiae* [108] were identified using the acyl-biotinyl exchange assay (ABE) that allows the specific substitution of the fatty-acyl chain by a thiol-reactive biotin derivative that is easily detectable with streptavidin. Studies to characterize the palmitoylated proteins in *S. cerevisiae* revealed that at least forty-seven proteins are palmitoylated [108]. The palmitoylated substrate proteins include the SNARE proteins (solid and N-ethylmaleimide sensitive factor attached receptors) involved in vesicles fusion [108] and heterotrimeric G proteins alpha subunits [116]. From the biochemical point of view, the best characterized palmitoyl transferase have in the active center a motif with the amino acids aspartic acid, two histidines and cysteine (DHHC), where the cysteine is the site for the S-palmitoylation. The DHHC motif in PATs is embedded in a 51-amino acids cysteine-rich region [117,118,119] related to the zinc finger transcriptional factors [120]. Some PATs, in yeast, e.g., Akr1 and Erf2-Shr5, are self-palmitoylated and the palmitic group in the acyl PAT is then transferred to the substrate protein [120]. Proteins containing the DHHC domain form a small family of otherwise heterogeneous proteins. Yeast studies confirm that the palmitic acid group attached to proteins is a hydrophobic core that serves to anchor proteins to either vesicles, organelles or plasma membranes as suggested previously [121,122]. Although the best characterized PATs contain the DHHC motif, it is important to note that palmitoylation has been reported in proteins that do not contain the DHHC domain [123]. A cysteine residue in the protein near a transmembrane domain may be sufficient for palmitoylation at that site. In this case, the transmembrane domain may anchor the PAT, allowing the palmitoylation of the close cysteine residue.

The yeast palmitoylome analysis reveals palmitoylated proteins with various physiological functions; of particular relevance are G-alpha proteins, two membrane phosphatases, the inositol-4 phosphate kinase and SNARE proteins, that are mediators of vesicles fusion. An important finding, in the context of vesicle-mediated protein secretion and vesicle fusion, was the observation that eight of the twenty-three SNARE proteins are palmitoylated. These eight proteins contain a cysteine close to a transmembrane domain, whereas other SNARE proteins that lack this fatty acyl acceptor domain are not palmitoylated [108]. The palmitoylation reaction appears to be important to target SNARE proteins to their plasma membrane location; alternatively, the palmitoylated SNARE proteins may be integrated into vesicle membranes, contributing to their fusion with the plasma membrane. A group of proteins that are palmitoylated in yeast are amino acid permeases with different substrate specificities. Noteworthy, these amino acid permeases contain 12 transmembrane domains, as occurs also with transporters involved in the traffic of penicillin and cephalosporin biosynthetic intermediates [124], although no studies have been made on the palmitoylation of the penicillin and cephalosporin C peroxisomal or vacuolal transporters. Interestingly, eleven of these amino acid permeases contain a cysteine residue in a conserved amino acid sequence (Phe-Trp-Cys) near the C-terminal end that is essential for palmitoylation of these proteins [108]. In studies of mutants defective in every PAT, a certain degree of substrate specificity was observed; for example, in *F. oxysporum*, the palmitoylation of the amino acid permeases is performed by the dedicated palmitoyl transferase Pfa4. Similarly, the SNARE proteins are known to be palmitoylated by the Swf1 palmitoyl transferase [108,125]. It is still unclear how frequent overlapping palmitoylation of some protein substrates by two or more PATs [126].

### 4.2. Palmitoyl Transferases in Filamentous Fungi

Less information is available on palmitoyl transferases in filamentous fungi despite the interest of protein S-acylation in fungal metabolism. The *A nidulans* Akr1 palmitoyl transferase consist of 737 amino acids that include five predicted transmembrane domains, six ankyrin repeats in the N terminal region and the DHHC motif between transmembrane domains 3 and 4 [127]. Green fluorescent protein labelled Akr1 co-localizes with Golgi markers, suggesting that palmitoylation of substrate proteins occurs in this organelle [127]. A detailed characterization of the *A. nidulans* Akr1 protein shows that this protein is self-palmitoylated in *S cerevisiae* [127]. Proteins with similar size (730 to 750 amino acids) with high similarity to Akr1 occur in many ascomycetes, e.g., *P. chrysogenum* has 731 amino acids (69.7% identity). Importantly a calcineurin-dependent regulatory element (CDRE) was found upstream of the *A. nidulans akr1* gene, and transcription of this gene was shown to be under calcium-calcineurin regulation (see Section 6). 

Studies on the palmitoyl transferases of *A. nidulans* and *C. neoformans* reveal that they are involved in cell wall stress and in the response of these fungi to metal ions [118,119,127,128,129]. 

#### 4.2.1. Palmitoylation of the RAS GTPase in Basidiomycetes and Ascomycetes

Detailed studies have been made on the palmitoylation of the RAS protein in *A. fumigatus* and in the basidiomycete *C. neoformans* [[16],,[128],[130]]. The RAS protein is a conserved GTPase; in yeasts and filamentous fungi, the activity of this protein is tightly regulated by prenylation and palmitoylation in the C-terminal region. In the human pathogen *C. neoformans*, the DHHC class palmitoyl transferase Pfa4 gene is required for acylation of the RAS protein and for its membrane localization. Mutants of *C. neoformans* defective in Pfa4 showed impaired growth and decreased virulence [118].

Similar studies of palmitoylation of the *A. fumigatus* RAS protein indicate that palmitoylation by Paf4 is required for targeting RAS to the membrane of actively growing hyphae [130]. Inhibition of the palmitoylation reaction by the palmitic acid analogue 2 Brome palmitate shows a reduction in the growth polarity and decrease of the pathogenicity of *A. fumigatus*. These results were confirmed with mutants defective in the Paf4 PAT [130,131]. 

#### 4.2.2. Protein Palmitoylation in Plant Pathogenic Fungi: The Cargo Adaptor Protein Complex

Recent studies on the *F. oxysporum* subspecies *niveum*, the cause of watermelon wilt, have revealed in this fungus a total of 211 proteins that are palmitoylated. Some of these palmitoylated proteins were found to be involved in pathogenicity and virulence mechanisms as reported previously in the filamentous fungi *F. graminearum, B. cinerea*, *Verticillium dahliae* and *Aspergillus flavus* [,[132],[133]]. *F. oxysporum* has six PAT genes (*pat1* to *pat6*) and the six PAT proteins contain the characteristic DHHC motif. The six palmitoyl transferase genes were cloned and disrupted in the parental strain. Using the ABE assay, it was confirmed in vitro that the product of three of the PAT genes (*pat1*, *pat2* and *pat4*) encodes enzymes with palmitoyl transferase activity, and it was shown that the three PATs are self-palmitoylated, whereas this was not confirmed in the other three PAT gene products [134]. Directed mutation of *pat1*, *pat2* and *pat4* showed that their palmitoyl transferase activity is dependent upon the presence of a cysteine residue in the DHHC motif. These three palmitoyl transferases have clear effects on growth, formation of asexual spores and conidia morphology, in addition to the response to the cell wall and metal ions stressing factors [135]. An important observation was the fact that the lack of Pat1, Pat2 or Pat4 reduces fungal virulence in plants probably by alteration of the formation of toxic metabolites [106,136].

*F. oxysporum* palmitoylated proteins include components of the cargo adaptor protein complex [135]. The cargo adaptor protein complex (AP-2) consists of four subunits, A, B, M and S, that play key roles in self-differentiation and pathogenicity [137]. Due to of the high relevance of the cargo adaptor complex in metabolism, this complex has been investigated in *S. cerevisiae*, *Candida albicans* and in several filamentous fungi. In yeast, this complex is important in polarity growth and cell wall integrity [129,138,139]. 

Studies in *Aspergillus* species indicate that the AP-2 complex plays an essential role in growth and apical localization of the membrane endocytic process [140]. Similar observations have been made in the wheat scab pathogen *F. graminearum* hyphae [132]. The cargo adaptor protein of this fungus (FgAP-2) plays an important role in the initial steps of the infection process and the complex subunits are involved in pathogenicity. The FgAP-2 complex is localized in the collar region of the hyphae tips and is involved in the activity of the lipid flippases during endocytosis. In conclusion, the available evidence indicates that the PATs of *F. oxysporum* subspecies *nivium*, *F. graminearum* and other fungi play an essential role in modification of the components of the cargo protein adaptor complex that is involved in regulation of growth, differentiation, response to stressing factors, protein traffic and virulence. Further studies are required to identify which of these mechanisms are key in the determination of the fungal virulence. 

## 5. Targeting of Specialized Metabolites Biosynthetic Enzymes to Vesicles/Endosomes by Posttranslational Palmitoylation

An important step forward to understand the melanin secretion mechanism was the finding that the four early enzymes of the melanin pathway are S-palmitoylated. Palmitoylation studies in *A. fumigatus* revealed that it contains two hundred thirty-four putative palmitoylated proteins, ninety-nine of which have been confirmed experimentally. In addition to the early melanin biosynthesis enzymes, the well-known RAS GTPase and other G proteins were found to be palmitoylated [130,131]. The key polyketide synthase, Alb1, is highly palmitoylated in all stages of the fungal growth, while palmitoylation of the trimming down enzyme Ayg1 and the reductases ArpR1 and ArpR2 takes place only during conidiation, suggesting that the palmitoylation modification is dependent upon conidiation signals [23]. The palmitoylation of these key melanin biosynthesis enzymes was confirmed by three methods: (1) using the biotin-based ABE assay, (2) by inhibition of the palmitoylation reaction with the palmitic acid structural analogue 2-brome-palmitate using the palmitoylation of the RAS protein as control and (3) by LC-MS/MS, which allows reliable identification of the palmitoylated substrate protein [22,141]. Palmitoylation of the early enzymes of the melanin pathway is critical for their localization in endosomes [23]. The finding of palmitoylation of the four early enzymes of melanin biosynthesis raises a new perspective on the mechanism for targeting enzymes involved in the biosynthesis of specialized metabolites and opens the question of whether palmitoylation of biosynthetic enzymes is a common mechanism in the secretion of specialized metabolites or if it is involved in the secretion of only some of the fungal products. However, when these palmitoylated enzymes are directed to peroxisomes and later to endosomes, their putative migration mechanism is still unclear (Figure 2). Six enzymes involved in the biosynthesis of three other *A. fumigatus* specialized metabolites, namely endocrosin, gliotoxin and fumitremorgin, tagged with GFP, were found to be located in vesicles, although it is unclear if they are palmitoylated. Several polyketide synthases and non-ribosomal peptide synthetases of *A. fumigatus* were predicted bioinformatically to be also palmitoylated enzymes [23]; however, additional experimental work is needed to confirm that the palmitoylation of the PKSs or NRPSs is required for targeting them to endosomes or other different secretion vesicles. 

Another group of substrate proteins that are palmitoylated by Akr1 in *A. nidulans* are three enzymes involved in ergosterol biosynthesis, namely Erg11A, Erg11B and Erg5. Palmitoylation of these enzymes may affect the biosynthesis of these sterols and also that of isoprenoids in fungi that are synthesized in a pathway similar to that of sterols, e.g., clavaric acid produced by the basidiomycete *Hypholoma sublateritium* [142,143]. The enzymes involved in clavaric acid biosynthesis have been characterized but there are no studies on the palmitoylation of these enzymes. The sterols are very important components in the membrane as determinants of the resistance to antifungal agents; therefore, the palmitoylation of these Erg proteins may have an important role in the resistance to azoles in filamentous fungi. Azoles trigger the membrane stress response that results in an increased flux of calcium and activation of the calcium homeostasis mechanisms. Importantly, this membrane stress response is abolished in Akr1-defective mutants, suggesting that there is an involvement of palmitoylation in this response. Similarly, the endoplasmic reticulum stress response to salts or antifungal agents, such as tunicamycin, is also absent in the *A. nidulans akr1* mutant [127], supporting the conclusion that the Akr1 palmitoyl transferase plays an essential role in the regulation of calcium metabolism in filamentous fungi. A summary of the characteristic features of the palmitoylation reactions in different yeasts and filamentous fungi is listed in Table 3.

## 6. Connection between Protein Palmitoylation and the Calcium/Calcineurin Regulatory Cascade

The calmodulin/calcineurin signaling cascade plays a crucial role in the control of fungal metabolism. Both primary and secondary metabolism respond to calcium levels in the culture broth, e.g., in response to calcium limitation and to different stressing factors [145]. Importantly, the palmitoyl transferase Akr1 of *A. nidulans* is involved in the control of calcium homeostasis in this filamentous fungus [127]. In recent decades, significant advances have been made on the transport of calcium into fungal cells and on the calcium-mediated regulation of primary and secondary metabolism. Calcium is introduced into fungal cells by several mechanism that are classified in high affinity influx systems (*hax*) and low affinity transporters (*lax*) [145]. The cytosolic calcium content is maintained by control of the influx from the extracellular medium by calcium pumps, calcium channels and calcium antiporters, and by release of calcium into the cytosol from store organelles [145,146,147,148,149]. In calcium starvation conditions or when the cell membrane or the endoplasmic reticulum are under stress by chemicals in the environment, the fungal cells react, stimulating the *hax* system; this increases the cytosolic calcium content and the resulting calcium transient increment triggers the calcium–calcineurin cascade [149]. The calcium-responsive zinc finger transcriptional factor CrzA is then phosphorylated and migrates from the cytosol to the nucleus where it recognizes and binds specific calcineurin-dependent regulatory elements (CDRE), which activates the cell response to balance calcium levels [150,151]. Deletion of the *akr1* gene resulted in slow growing colonies, as occurs also in mutants defective in the Hax calcium transporter system [127]. Furthermore, these authors found that there is a CDRE motif in the upstream region of the *A. nidulans akr1* gene; therefore, the palmitoyl transferase Akr1 is involved in calcium homeostasis in this fungus and its expression is under the control of calcium by the binding of calcineurin-responsive CrzA transcriptional factor to the CDRE motif [127]. Mutants defective in Akr1 are extremely sensitive to calcium chelating agents such as EGTA. The Akr1 protein labelled with the enhancer green fluorescent protein was shown to co-localize with late Golgi markers [152,153]. As indicated above, the Akr1 protein of *A. nidulans* contains the characteristic ^484^DHHC^487^ motif located between transmembrane domains 3 and 4. Characterization of a truncated mutant that lacks the DHHC motif and a different strain carrying a mutation of the cysteine^487^ to serine confirmed that the DHHC motif is required for the palmitoylation effect on calcium homeostasis. Comparative analysis of the palmitoylated proteins in the parental strain and in an *akr1* defective mutant showed that several proteins were substrates of the Akr1 palmitoyl transferase; some of these proteins were involved in membrane phosphate traffic and calcium homeostasis (e.g., two P-type ATPases, a putative V type proton ATPase and three proteins known to be induced by calcium stress).

### Palmitoylation of Calcineurin Targets This Phosphatase to the Golgi and Plasma Membrane

An important finding in the context of calcium regulation is the observation that a isoform of the calcineurin, named CNAβ1, is modified by palmitoylation in mammals [154]. In contrast to the non-palmitoylated wild type calcineurin that is located in the cytosol, the CNAβ1 calcineurin isoform is palmitoylated in two positions in the divergent C-terminal tail. The calcineurin palmitoylated form is located in the Golgi and plasma membranes, where it interacts with different sets of proteins including the inositol-4 kinase complex; this complex plays an important role in the control of the calcineurin activity and the calcium signaling cascade by phosphorylation [145]. Palmitoylated forms of calcineurin have been reported in several mammalian cells [155,156] but the presence of a similar calcineurin form in fungi has not been studied so far. In summary, the localization of the palmitoylated calcineurin in the Golgi membrane allows interaction of calcineurin with a set of proteins different from those recognized by calcineurin in the cytosol.

## 7. Conclusions and Future Outlook

In recent decades, significant advances have been made on the knowledge of the subcellular localization of biosynthetic enzymes of specialized metabolites in filamentous fungi. The localization of these enzymes is very diverse [92,157]; many of these enzymes are located in the cytosol following the formation by the protein synthesis machinery and polyribosomes; subsequently, many of these enzymes are targeted to peroxisomes, endosomes, the endoplasmic reticulum, internal membrane systems or the plasma membrane and even to the cell wall. The early observation that PKSs and NRPSs were located in the cytosol has now been revised, suggesting that these large enzymes are associated with vacuole membranes or located in vesicles [21,49]. Several examples indicate that the last enzymes of different secondary metabolite biosynthetic pathways are associated with the cell wall or are completely extracellular. This is the case of the last enzyme(s) involved in the biosynthesis of fumiquinazoline C and in the formation of melanin in at least two different fungi *A. fumigatus* and *B. cinerea*. In this last fungus, the second half of the biosynthetic pathway, from scytalone to melanin, has been shown to be entirely extracellular [21]. The extracellular localization of the final steps of some biosynthetic pathways is related to the toxicity of some late intermediates or the final product and serves to protect the producer cell against the toxicity of those products. Changes in the subcellular localization of precursors, biosynthetic intermediates and the final products highlight the need for elaborated control mechanisms that coordinate the transfer of the biosynthetic enzymes and their products from one organelle to another and finally to the extracellular space. Advances have been made recently in our understanding of the traffic mechanisms of specialized metabolites biosynthetic enzymes from the cytosol or peroxisomes into vesicles, particularly endosomes. However, the final stages of the secretion from endosomes or related vesicles to the extracellular space is still unclear. Initial evidence has been obtained indicating that endosomes are fused with the plasma membrane, thus releasing the final products in the cell wall [39,55,56]. This is supported by the finding that many of the MFS or ABC transporters are located in the intracellular membranes or in the plasma membrane, facilitating the traffic of intermediates between organelles or the efflux to the extracellular space. Importantly, posttranslational modification of proteins, particularly palmitoylation by PATs, targets the modified protein to internal membrane systems or to the plasma membrane. Notable advances have been made on the elucidation of palmitoylation mechanisms by palmitoyl transferases both in yeast and filamentous fungi [158]. The palmitoylation of the first enzymes of the biosynthesis of melanin include the Alb1 PKS, which is palmitoylated during the growth and sporulation of *A. fumigatus*, whereas the subsequent enzymes Ayg1 and the reductases Arp1 and Arpr2 are only palmitoylated following induction of conidiation [23]. These posttranslational modifications are required to target the enzymes to endosomes. Six other *A. fumigatus* specialized metabolites biosynthetic enzymes involved in the formation of endocrosin, gliotoxin and fumitremorgin were also located in vesicles, but it is unclear if they are palmitoylated. Bioinformatic analysis predicts that several NRPSs and PKSs of *A. fumigatus* may be palmitoylated, but more experimental work is required to confirm this prediction. The hydrophobic palmitoyl tail serves to attach the palmitoylated protein to the cell membranes. The hydrophobic chain is removed by dedicated depalmitoylases or by non-specific thiosterases when required. The conditions and the timing for the removal of these tails need to be studied in more detail. Finally, strong evidence indicates that fungal PATs of the DHHC type are involved in the palmitoylation of enzymes involved in calcium metabolism, affecting calcium transport and homeostasis mediated by the phosphatase calcineurin. In higher eukaryotic cells, an isoform of calcineurin is palmitoylated and appears to be targeted to membrane systems, thus facilitating interaction of this calcineurin form with membrane proteins, affecting, therefore, calcium transport and metabolism. Palmitoyl transferases different from the DHHC type are known to exist in filamentous fungi but they are very poorly studied; investigation on the scope of posttranslational modifications of biosynthetic enzymes needs to be expanded. Basidiomycetes fungi also produce a significant number of secondary metabolites but the subcellular localization of their biosynthetic enzymes is largely unknown [5,159,160]. Only in the case of the basidiomycete *C neoformans* is there good knowledge of the palmitoylation of some substrate proteins, including the RAS GTPase, as described above [130,131] and, therefore, it is likely that similar palmitoylation reactions occur in other basidiomycetes; more studies are required in this field. In summary, the novel information on the posttranslational palmitoylation of biosynthetic enzymes is scientifically challenging and opens new avenues for understanding the traffic and secretion of bioactive metabolites in the near future.

## Figures and Tables

**Figure 1 ijms-25-01224-f001:**
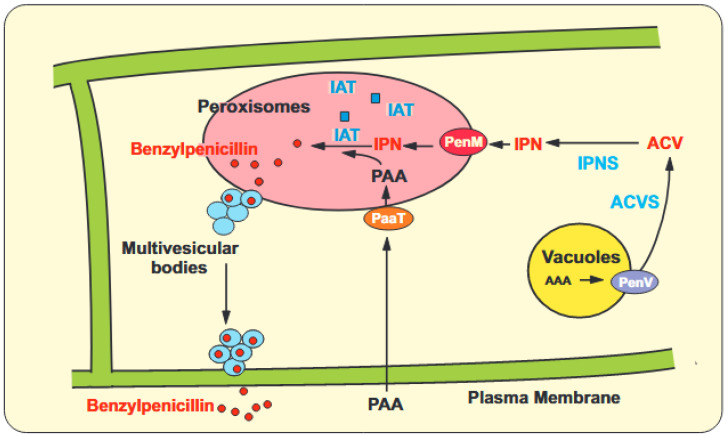
Model of compartmentalization of the penicillin biosynthesis pathway in *Penicillium chrysogenum*. A peroxisome is shown by a large pink ellipse. Both Isopenicillin N (IPN) and Phenylacetic acid (PAA) are introduced into peroxisomes by the PenM (red ellipse) and PaaT (orange ellipse) transporters, respectively. The vacuoles (yellow circle) provide α-aminoadipic acid through the PenV transporter (purple ellipse). In the cytosol, the ACV synthetase combines α-aminoadipic acid, cysteine and valine to form the tripeptide ACV, which is cyclized to IPN by the IPN synthase. The release of α-aminoadipic acid from IPN (to form 6-APA) and the acylation of 6-APA by the isopenicillin N acyltransferase (IAT) (blue squares), that results in benzylpenicillin formation (red circles), occur in peroxisomes. The benzylpenicillin produced is transported to multivesicular bodies (blue circles) and finally secreted to the external medium by exocytosis. The organelles are not drawn at scale.

**Figure 2 ijms-25-01224-f002:**
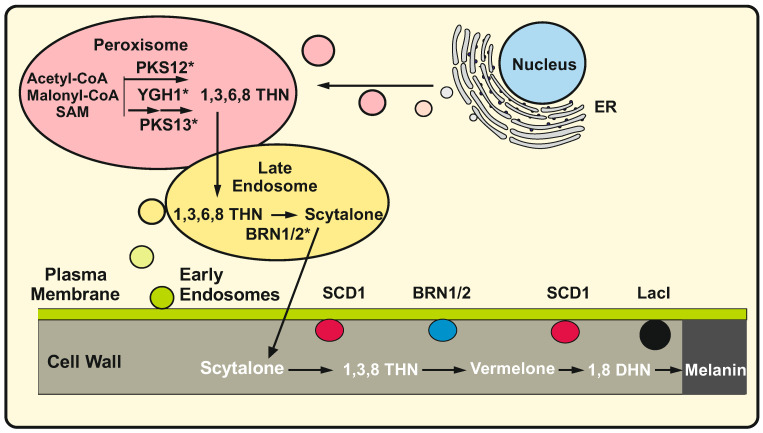
Biosynthesis of melanin in *Botrytis cinerea.* The palmitoylated enzymes are labelled with an asterisk. In peroxisomes (pink ellipse) the BcPKS12/13 polyketide synthases together with the trimming enzyme BcYGH1 combine Acetyl CoA and Malonyl-CoA to form 1,3,6,8 THN. This compound is transported to Late endosomes (yellow ellipse) and converted in scytalone by the BcBRN1/2 reductases. Scytalone is excreted to the cell wall space where the SCD1 enzyme (red circles) forms 1,3,8 Trihydroxynaphtalene (1,3,8 THN). This compound is reduced to vermelone by reductases BRN1/2 (blue circle) which are located both in endosomes and the cell wall and reduced again by the SCD1 to 1,8 DHN. DHN polymerizes with the help of the laccase LacI (black circle) to melanin which is deposited in the cell wall (see text for details).

**Table 1 ijms-25-01224-t001:** Documented examples of localization of secondary metabolites biosynthetic enzymes in peroxisomes.

Producer Fungus	Final Product	Enzyme (s) in the Pathway	Supporting Evidence	References
*Penicillium chrysogenum* *Aspergillus nidulans*	Penicillin	Isopenicillin N acyltransferase Phenylacetyl-CoA ligase	Immunoelectron microscopy PTS1 sequences in both enzymes Peroxisome-less mutants Located in purified peroxisomes	[8,12,13]
*Acremonium chrysogenum*	Cephalosporin C	Isopenicillin N-CoA ligase Isopenicillinyl N-CoA epimerase	PTS1 targeting sequences	[8,14]
*Aspergillus parasiticus* *Aspergillus flavus*	Aflatoxin	AflA, B, C (PKS complex) HypC: anthrone oxidase AflJ (early times)	Norsolorinic acid accumulation PTS1 sequences	[15,16]
*Aspergillus nidulans*	Sterigmatocystin	AflA, B, C	Accumulation of norsolorinic acid	[15]
*Alternaria alternata*	AK toxins	AK1: carboxyl activating enzyme AK2: α,β hydrolase AK3: Enoyl-CoA hydratase	PTS1 targeting sequence Peroxisome deficient mutants Fluorescent labelled enzymes	[9,17,18,19]
*Penicillium brevicompactum*	Mycophenolic acid	PbACL891: Acyl-CoA ligase MpaH’: Acyl-CoA hydrolase	Fluorescent labelled enzymes PTS1 targeting sequences	[20]
*Aspergillus fumigatus* *Botrytis cinerea*	Melanin	BcPKS12: Sclerotia polyketide synthase BcPKS13: Conidia polyketide synthase BcYGH1 PK trimming down hydrolase	PTS1 sequences Fluorescence labelled enzymes	[21,22,23]
*Penicillium patxilli*	Patxilin	PaxG: Geranyl-Geranyl-PP synthase B	PTS1 targeting sequence Patxilin negative mutants Fluorescent labelled enzymes	[24]
*Aspergillus fumigatus*	Fusarinine	SidI: mevalonyl-CoA ligase SidH: mevalonyl-CoA dehydratase SidF: Anhidromevalonyl-CoA transferase	C-terminal PTS1 (SidH, SidF) N-terminal PTS2 (SidI)	[25]

**Table 2 ijms-25-01224-t002:** Documented localization of secondary metabolite biosynthesis enzymes ^1^ in (A) Vesicles or Endosomes, (B) Cell wall space.

Producer Fungus	Final Product	Enzyme (s) in the Pathway	Supporting Evidence	References
**A. Vesicles or Endosomes**				
*Aspergillus parasiticus* *Aspergillus flavus*	Aflatoxins	AflM (Ver-1): NADPH-dep. reductase AflD-(Nor-1): NADPH-dep. ketoreductase. AflP (OmtA): Methyl transferase AflJ: Transcriptional Co-activator (late times)	Fluorescence labelled enzymes	[16,26,27,28]
*Fusarium graminearum*	Trichothecenes	Hmr1: Hydroximethylglutaryl-CoA reductase Tri5: Trichodiene synthase Tri1: Calonectrin oxygenase Tri12: transporter	Tri12 mutants Fluorescence labelled enzymes	[29,30,31]
*Aspergillus fumigatus*	Fumiquilazoline C	FmqA: Trimodular NRPS FmqE: Transporter	Fluorescent labelled enzyme Microscopic vacuole stain	[32,33,34]
**B. Cell Wall**				
*Aspergillus fumigatus*	Fumiquinazoline C	FMQD: Fimiquinazoline oxidoreductase	Fluorescent labelled enzyme	[4,34]
*Aspergillus fumigatus*	Melanin	Abr1, Abr2 Laccases ^3^	Fluorescent Labelled enzyme	[23]
*Botrytis cinerea*	Melanin	BcSCD1: Scytolone dehydratase BcBRN1/2: Trihydroxynaphthalene reductase ^2^ LaccaseI ^3^	Fluorescent labelled enzyme	[21]

^1^. Some enzymes located in peroxisomes are shown in Table 1 and when they migrate to vesicles, are also included in Table 2 (see text for details). ^2^. The BcBRN1 plays a duplicated role, intracellular and extracellular (see text for details). ^3^. The laccases are secreted by the conventional protein secretory pathway. Do not require palmitoylation.

**Table 3 ijms-25-01224-t003:** Important features of yeasts and filamentous fungi palmitoyl transferases.

Yeast/Fungi	Important Features	References
**Yeasts**		
*Saccharomyces* *cerevisiae*	1. *S. cerevisiae* contains seven palmitoyl transferases. 2. Some PATs are self-palmitoylated before transferring the acyl group to other substrate proteins. 3. The best-known PATs contain a DHHC motif essential for their activity. The cysteine is the site for S-acylation. 4. A cysteine near a transmembrane domain is the site for S-acylation in PATs that lack the DHHC motif. 5. At least 47 palmitoylated proteins are known. 6. The palmitoylated proteins include heterotrimeric G proteins alpha subunits, SNARE proteins, amino acid permeases, membrane phosphatases, Inositol-4-phosphate kinase. 7. The SNARE proteins and the amino acid permeases are acylated by dedicated PATs.	[108,115,116,117,118,119,120,121,122,123,125,128]
**Fungi: Ascomycetes**		
*Aspergillus nidulans*	1.The model PAT Akr1 contains 737 amino acids a DHHC motif and 5 transmembrane domains and is well conserved in Ascomycetes. 2. Akr1 is located in the Golgi system. 3. Akr1 is self-palmitoylated. 4. The *akr1* gene is regulated by calcium/calmodulin through a calcineurin-dependent regulatory element located upstream of the gene.	[120,127,144]
*Aspergillus fumigatus*	1. *A. fumigatus* contains 234 palmitoylated proteins, 99 of them fully confirmed. 2. The Akr1 PAT acylates four melanin biosynthetic enzymes. 3. Palmitoylation of three of the melanin biosynthetic enzymes is triggered by conidiation inducing signals. 4. Palmitoylation of two early enzymes of the melanin pathway is essential for their localization in endosomes. 5. PAT4 palmitoylates the RAS protein. 6. The PAT4 defective mutants show reduced growth polarity and decreased pathogenicity.	[22,23,122,130,131]
*Fusarium oxysporum* var *niveum*	1. *F. oxysporum* var. *niveum* contains six palmitoyl transferases (PAT1 to PAT6) and 211 palmitoylated proteins. 2. The six PATs contain the DHHC motif. Only PAT1, 2 and 4 show in vitro PAT activity, and are self-palmitoylated. 3. Palmitoylation affects growth, differentiation, cell wall stress and sensitivity to metal ions stress. 4. PAT mutants show reduced virulence in watermelon plants. 5. The palmitoylated proteins include components of the cargo adaptor protein complex (AP-2) which plays a key role in the localization of the membrane endocytic process in hyphae tips.	[134],[135]]
*Fusarium graminearum*	1. Component proteins of the cargo adaptor FgAP-2 are palmitoylated. 2. The FgAP-2 complex regulates growth, polarity and apical localization of lipid flippases during endocytosis. 3. Palmitoylation of the FgAP-2 components plays important roles in the early stages of pathogenicity in wheat infection.	[132]
**Fungi: Basidiomycetes**		
*Cryptococcus neoformans*	1. *C. neoformans* has seven palmitoyl transferases. 2. The PATs are not essential for growth, although mutants in PAT3 and PAT 4 show distinct degrees of temperature sensitivity. 3. PAT4 is involved in acylation of the RAS GTPase. 4. Palmitoylation of the RAS protein is required for its proper localization in the cell membrane. 5. Mutants in the RAS protein show pronounced morphological defects in both the yeast and mycelial form. 6. Palmitoylation by PAT4 affects the pathogenicity in humans.	[118,119,131,132]

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
