# Peer review of "Targeting of Specialized Metabolites Biosynthetic Enzymes to Membranes and Vesicles by Posttranslational Palmitoylation: A Mechanism of Non-Conventional Traffic and Secretion of Fungal Metabolites"

_ijms, 2024, doi:10.3390/ijms25021224_

Round 1

Reviewer 1 Report

Comments and Suggestions for Authors

This is an interesting revision article that grouped information concerning the posttranslational modification of fungal proteins with enzymatic activity involved in different biochemical pathways of the secondary metabolites biosynthesis. For me, the document can be understood as two main parts. The first one is on the description of the biosynthetic pathways of the different secondary metabolites, characterizing the enzymes, gene clusters, precursors and substrates. Another is the specific presentation of the S-acylation of proteins through palmitoylation and its impact on the location and movement of the key enzymes in the biosynthetic pathways of the secondary metabolites. In spite of the excellent presentation of the knowledge on this thematic, it is also clear that there are many gaps and questions to be investigated on these important biological events in fungi. It is clear the importance of the palmitoylation event.

The document is well written and moderate points need revision to improve the English language. Tables and figures are important to help the understanding of information presented. Considering that the article is dense considering the different pathways described, the figures have central importance. So, the figure 2 needs improvement. I believe that this figure does not reflect all information presented in the text.  The SCD1, BRN1/2 and Lac1, enzymes involved in the conversion of scytalone to melanin, are presented in the cell membrane (figure 2). However it was mentioned that this conversion takes place in the cell wall space (line 429). Additionally, it was also mentioned that the SCD1 and BRN12 are associated enzymes to the cell wall (lines 435-439). As presented, apparently the conversion is totally outside the cell wall space. Clarification on this aspect is necessary. The action of these enzymes is not clear in the figure, in spite of the description presented in the legend.

The description of the biosynthetic pathways was directed mainly for ascomycetes (see also table 1 and 2). If the idea of the revision is to gather information on traffic and secretion of fungal specialized metabolites as general aspects, considering the posttranslational modified enzymes, information on basidiomycetes also could be included. For example, it was mentioned only the sexual ascospores (line 54-55).

Minor:

Line 105-106 – I believe that the reference was lost. Please add.

Line 198-199 – Replace “in the biosynthesis of AK toxin” by “in this biosynthetic pathway”.

Line 229 – The sidF gene should be italicized.

Line 218 – Correct “fusarinin” to “fusarinine”.

Line 298 – The “Penicillium” should be italicized.

Line 521 – Correct “denin” to “dinein”.

In general, verify the construction of sentences and the period in the sentences.      

Comments on the Quality of English Language

Moderate suitability necessary.

Author Response

Reviewer 1

Thank you for your positive evaluation of our work

The document is well written and moderate points need revision to improve the English language. Tables and figures are important to help the understanding of information presented. Considering that the article is dense considering the different pathways described, the figures have central importance. So, the figure 2 needs improvement. I believe that this figure does not reflect all information presented in the text. The SCD1, BRN1/2 and Lac1, enzymes involved in the conversion of scytalone to melanin, are presented in the cell membrane (figure 2). However, it was mentioned that this conversion takes place in the cell wall space (line 429). Additionally, it was also mentioned that the SCD1 and BRN12 are associated enzymes to the cell wall (lines 435-439). As presented, apparently the conversion is totally outside the cell wall space. Clarification on this aspect is necessary. The action of these enzymes is not clear in the figure, in spite of the description presented in the legend.

Answer: The reviewer is right. The text and the figure legend are correct but unfortunately the localization of the enzymes in Figure 2 was unclear. Figure 2 has been corrected and both the above mentioned enzymes and the corresponding late steps of the pathway have been inserted in the cell wall.

The description of the biosynthetic pathways was directed mainly for ascomycetes (see also table 1 and 2). If the idea of the revision is to gather information on traffic and secretion of fungal specialized metabolites as general aspects, considering the posttranslational modified enzymes, information on basidiomycetes also could be included. For example, it was mentioned only the sexual ascospores (line 54-55).

Answer: This article is focused on the description of those filamentous fungi in which there are solid evidence of the localization/palmitoylaton on secondary metabolites biosynthetic enzymes. Most of them are Ascomycetes but in the text, there was also a paragraph regarding the basidiomycete C. neoformans (lines 626-633) now as a new subsection 4.2.1. The basidiomycetes nature of C. neoformans has been highlighted modifying the sentence as follows “the basidiomycete C. neoformans”. In lines 54-55 we have modified the text as follows “with the formation of sclerotia, asexual conidia, sexual ascospores or basidiomycetes fruiting bodies” and added the reference (5) of Baranda-Herath et al. on basidiomycetes secondary metabolites. In addition, we have included new sentences on the Future Outlook section about the development of secondary metabolites in basidiomycetes (lines 832-837) including the references 161 and 162.

Minor:

Line 105-106 – I believe that the reference was lost. Please add.

Answer: The sentence that you mention does not describe a single article but refers to the entire content of section 4. Therefore, rather than a citation we have included the sentence “(see section 4 below)” refering to the whole S-acylation content of section 4

Line 198-199 – Replace “in the biosynthesis of AK toxin” by “in this biosynthetic pathway”.

Answer: Corrected as suggested in line 199

Line 229 – The sidF gene should be italicized.

Answer: Corrected as suggested

Line 218 – Correct “fusarinin” to “fusarinine”.

Answer: Corrected as suggested

Line 298 – The “Penicillium” should be italicized.

 Answer: Corrected as suggested

Line 521 – Correct “denin” to “dinein”.

 Answer : Corrected to dynein in line 523

In general, verify the construction of sentences and the period in the sentences.

Answer: the English has been carefully revised, including the periods and corrected when necessary

Reviewer 2 Report

Comments and Suggestions for Authors

The manuscript by Martín and Liras concerns a difficult issue related to alternative protein transport routes and their modifications that ensure this transport to specific cellular locations. This is an interesting and important observation and summary. The work is very detailed, clearly written and describes the topic thoroughly.

The minor comment is that the authors should consider whether in the second part of the manuscript they could also summarize the information in the form of a table or figure, which would be helpful in illustrating the subject.

Moreover, in the last subsection CHHD was written twice instead of DHHC.

The text also contains a few typos, unnecessary dots and minor grammatical errors that should be corrected, e.g. in Table 2 (Cell wall), and in lines 64, 109, 324, 521, 523, 565, 566, 715, 753.

Comments on the Quality of English Language

 There are some minor grammatical mistakes that should be corrected.

Author Response

Reviewer 2

Authors: Thank you for your positive suggestions

The minor comment is that the authors should consider whether in the second part of the manuscript they could also summarize the information in the form of a table or figure, which would be helpful in illustrating the subject.

Answer: Indeed, the content of the second part of the article is dense and therefore, as suggested by reviewer 2 we have prepared Table 3 describing the characteristic features of the yeast palmitoylation and filamentous fungi divided into Ascomycetes and Basidiomycetes (as suggested by reviewer 1). In principle this is entitled as Table 3 but if the editor prefer it may be entitled Box 1

Moreover, in the last subsection CHHD was written twice instead of DHHC.

Answer: Changed in lines 825 and 830

The text also contains a few typos, unnecessary dots and minor grammatical errors that should be corrected, e.g. in Table 2 (Cell wall), and in lines 64, 109, 324, 521, 523, 565, 566, 715, 753.

Answer: Corrections have been made in Table 2 and the previous lines 64, 109, 324, 521, 523, 565, 566, 715 as suggested. The sentence in line 753 has been revised carefully and we have included the specific proteins in a parenthesis in lines 763-764.